# Genistein Has Antiviral Activity against Herpes B Virus and Acts Synergistically with Antiviral Treatments to Reduce Effective Dose

**DOI:** 10.3390/v11060499

**Published:** 2019-05-31

**Authors:** Julia C. LeCher, Nga Diep, Peter W. Krug, Julia K. Hilliard

**Affiliations:** 1Department of Molecular and Cellular Biology, Kennesaw State University, Kennesaw, GA 30189, USA; 2Viral Immunology Center, Department of Biology, Georgia State University, Atlanta, GA 30303, USA; diepnga103@gmail.com (N.D.); pkrug@gsu.edu (P.W.K.); jhilliard@gsu.edu (J.K.H.)

**Keywords:** herpes B virus, macacine herpesvirus-1, genistein, flavonoids, acyclovir, ganciclovir, antiviral agents

## Abstract

Herpes B virus is a deadly zoonotic agent that can be transmitted to humans from the macaque monkey, an animal widely used in biomedical research. Currently, there is no cure for human B virus infection and treatments require a life-long daily regimen of antivirals, namely acyclovir and ganciclovir. Long-term antiviral treatments have been associated with significant debilitating side effects, thus, there is an ongoing search for alternative efficacious antiviral treatment. In this study, the antiviral activity of genistein was quantified against B virus in a primary cell culture model system. Genistein prevented plaque formation of B virus and reduced virus production with an IC_50_ value of 33 and 46 μM for human and macaque fibroblasts, respectively. Genistein did not interfere directly with viral entry, but instead targeted an event post-viral replication. Finally, we showed that genistein could be used at its IC_50_ concentration in conjunction with both acyclovir and ganciclovir to reduce their effective dose against B virus with a 93% and 99% reduction in IC_50_ values, respectively. The results presented here illuminate the therapeutic potential of genistein as an effective antiviral agent against B virus when used alone or in combination with current antiviral therapies.

## 1. Introduction

Herpes B virus (Macacine herpesvirus-1) is an alphaherpesvirus endemic in macaque monkeys, an animal widely used in biomedical research [1]. B virus is of significant interest because it is capable of potentially deadly human zoonosis and thus poses a major occupational hazard to researchers, veterinarians, and animal handlers. B virus can be spread to humans via direct contact with infectious macaque body fluids and tissues during bites, scratches or splashes. Pathogenesis typically occurs within two weeks post-infection, with symptoms ranging from influenza-like illness, headache, vesicular lesions, ataxia and/or ascending paralysis. In humans, B virus is highly neuroinvasive and can lead to severe neurologic complications, such as transverse myelitis, encephalitis and potentially death. Without timely diagnosis and proper treatment, the mortality rate for infected individuals is as high as 70% [1,2,3,4,5]. Currently, there is no cure for human B virus infection, only life-long antiviral treatments, and patients live under the constant threat of potential virus reactivation.

The survival rate following human B virus infection is 80% when antiviral treatments are started early in infection [4]. Acyclovir (ACV) or ganciclovir (GCV) are typically the first drugs given in most clinical cases of B virus infection [1]. Following infection, antivirals are administered intravenously to put the virus into a latent state; then patients will remain on an oral antiviral regimen throughout their lifetime to prevent virus reactivation. ACV and GCV are attractive antivirals against B virus because they enter the host cell in an inactive form and must be phosphorylated intracellularly by virus-encoded thymidine kinase to become active [6,7,8]. However, these antivirals have several limitations. ACV, for example, has poor solubility in water, a short plasma half-life, and low blood-brain barrier passage. Following oral administration of ACV, there is a 20% bioavailability of the drug in the plasma and even lower levels in the cerebrospinal fluid [9,10]. In addition, in vitro studies have shown that B virus is approximately 10-fold less sensitive to ACV than the closely related human herpessimplex virus-1 (HSV-1) [8]. Thus, ACV must be administered in significantly greater concentration to control the disease, especially when intravenous administration is required. Increased doses and long-term administration comes with increased risk of negative side-effects. ACV can form crystals in the renal tubules during clearance from the kidney and, consequently, patients on ACV treatment often suffer from neurotoxicity associated with impaired renal function [11,12,13]. Further, there are reports of resistance of HSV-1 to ACV when patients are infected with virus isolates that carry mutations in either the DNA polymerase and/or the thymidine kinase gene [14]. It is anticipated that similar problems will ultimately arise when using ACV as a prophylactic treatment for B virus infection. Clearly, it is critical that new drugs be identified to provide a safer and more effective intervention against primary and reactivated B virus infections.

Genistein is a flavonoid found in plants of the Fabaceae (bean) family and in soy-based foods. In plants, flavonoids serve many functions critical to plant physiology. In humans, they possess many pharmacologic properties and therapeutic potential, such as antioxidant, anti-microbial, anti-carcinogenic, and anti-inflammatory activity, and have been summarized in several reviews [15,16]. In 1999, the Food and Drug Administration approved the food labeling of health claims of soy-based foods in reducing the risk of coronary heart diseases [17]. Since 1998, there have been 65 Phase I-III clinical trials registered with the National Institute of Health assessing the biological benefits of genistein and its derivatives [18].

Genistein is a general protein kinase inhibitor with selectivity towards tyrosine residues. Tyrosine phosphorylation regulates many cellular processes, including cell proliferation and angiogenesis. Therefore, genistein has been widely recognized as an anticancer agent, though the idea of flavonoids acting as antivirals is far from novel. Genistein has been shown to have antiviral properties against several herpesviruses including HSV-1, cytomegalovirus, and bovine herpesvirus-1, as well as non-herpesviruses, including SV40, human papilloma virus, porcine reproductive and respiratory syndrome virus, African swine fever virus, and human immunodeficiency virus [19,20,21,22,23,24,25,26,27,28,29,30,31,32,33,34,35]. Given the similarities of B virus to other herpesvirus against which genistein exhibits antiviral activities, we hypothesized that genistein would exhibit antiviral activity against B virus and, further, that genistein could be used in conjunction with current antiviral treatments to reduce their effective dose and enhance their antiviral properties.

## 2. Materials and Methods

### 2.1. Cells, Media, and Virus

Human foreskin fibroblasts (HFFs; ATCC**^®^** CCD-1112SK™, Manassas, VA, USA), Rhesus macaque foreskin fibroblasts (RMF; isolated in house), and African Green Monkey kidney cells (Vero; ATCC**^®^** CCL-81™, Manassas, VA, USA) were used in this study. The media compositions employed were (1) HFF: minimum essential medium (MEM), 10% inactivated fetal bovine serum (ΔFBS; Atlas Biologicals F-0050-A, Fort Collins, CO, USA), 100 U/mL penicillin-streptomycin (pen-strep; Mediatech/Corning #30-002-CL Tweaksbury, MA, USA), 1% sodium pyruvate (Mediatech/Corning #25-000-CL Tweaksbury, MA, USA), 1% glutaMAX™ (Gibco # 35050-061), and 1% non-essential amino acids (Mediatech/Corning #25-025-CL Tweaksbury, MA, USA); (2) RMF: Dulbecco’s modified essential medium (DMEM; Sigma-Aldrich #D-6046, St. LouisMOUSA), 18% ΔFBS, and 100 U/mL pen-strep; and (3) Vero: DMEM, 10% ΔFBS, 100 U/mL pen-strep. For all experiments, cells were grown at 37 °C in a 95% O_2_, 5% CO_2_ incubator. B virus (laboratory strain E2490) was propagated in Vero cells. Virus stock titer was determined by standard plaque assay in Vero cells. Propagation and harvesting of B virus isolates were performed in the Georgia State University (GSU) BSL4 Laboratory in accordance with the guidelines of the 5th edition of Biosafety in Microbiological and Biomedical Laboratories.

### 2.2. Chemicals and Reagents

Stock solutions of acyclovir (ACV) and genistein were prepared at 100 mM in dimethyl sulfoxide (DMSO) and ganciclovir (GCV) were prepared at 40 mM in sterile water. Citrate buffer was prepared with 50 mM sodium citrate and 4 mM potassium chloride in sterile water, pH 3. Methylcellulose was prepared as a 1% solution in MEM. Crystal violet was prepared as a 0.5% solution in diH_2_O with 20% ethanol.

### 2.3. Trypan Blue Dye Exclusion Assay

Cells were grown to ~75% confluency in 48-well plates. For experiments using synchronized cells, cells were serum-starved for ~18 h prior to treatment. Cells were then treated with 0–100 μM genistein or 0.1% DMSO (control) and grown for 48 h. Cells were collected by trypsinization (0.25% trypsin-EDTA; Gibco #25200-056), stained with trypan blue dye (Gibco #15250-061) and counted on a hemocytometer. All experiments were run in triplicate. Cell viability was calculated as the percent of live cells divided by the total number of cells.

### 2.4. Cell Proliferation Assay

Cell proliferation was measured in two ways. First, cells were grown, treated, and collected as above in the Trypan Blue Dye Exclusion Assay and then counted to obtain the total number of cells. This number was compared to the starting number of cells at the time of seeding. Secondly, cells were assayed by MTS assay (Promega-CellTiter 96**^®^** AQueous one solution cell proliferation assay). Cells were seeded into 96-well plates at ~75% confluency, allowed to adhere for ~6 h, then serum-starved for ~18 h before genistein treatment (0–100 μM) or 0.1% DMSO (control) was added to cells in fresh serum-containing medium. Cells were grown for an additional 48–72 h, then assayed for metabolic activity following the manufacturer’s protocol. Metabolic activity was calculated by subtracting background EU values (cells with no MTS reagent) from total EU values, then setting the untreated group to 100% and comparing the treated and untreated groups. All experiments were run in triplicate.

### 2.5. Bromodeoxyuridine (BrdU) Incorporation Assay

Cells were grown to ~90% confluency on 8-well chamber well slides (NUNC™ Lab-Tek™), then serum-starved for ~18 h, after which genistein treatment (0–100 μM) or 0.1% DMSO (diluent control) was added to cells in serum-containing medium with BrdU labeling reagent (Invitrogen 00-0103, Carlsbad, CA, USA). Cells were incubated for 4 h, then fixed and stained for BrdU following the manufacturer’s protocol (ThermoFisher, Waltham, MA, USA). Region of interest (ROI)analysis was performed using Zeiss AxioVision rel 4.8 software.

### 2.6. DNA Isolation and RT-PCR

DNA was isolated from cells using the manufacturer’s protocol with slight modification. Viral DNA was detected by real-time PCR using a 6-carboxyfluorescein/tetramethylrhodamine (FAM/TAMRA)-labeled probe with primers to glycoprotein G (F 5′ 3′ R 5′ 3′). DNA isolated from Vero-infected cells was used as a positive control. Average C_T_ values were calculated from replicate groups then compared to the uninfected control group.

### 2.7. Indirect Virus Yield Assay

HFFs or RMFs were grown to confluency in 12-well plates, then infected with 150 PFU of B virus for 1 h in the presence of 0–100 μM genistein or DMSO control. After absorption, the media was removed and fresh media added. After 48 h, cells/supernatants were collected, serially diluted, and added to Veros grown to confluency in 12-well plates for standard plaque assay.

### 2.8. Plaque Reduction Assay

Cells were grown to confluency in 12-well plates then infected with 150 PFU of B virus for 1 h in the presence of 0–100 μM genistein or DMSO control. After absorption, the media was removed and cells overlaid with 0.5% methylcellulose with the indicated concentration of genistein. To assay the effects of the drugs on virus replication, the old methylcellulose was replaced with new methylcellulose containing 50 μM genistein at 0, 2, 4, 6, and 8 h post-infection. For combination plaque assays, infected cells were treated with 50 μM genistein and 0–100 μM ACV or 0–40 μM GCV, both during and after the absorption period. In all cases, at 48 h post-infection (hpi), cells were fixed with 100% methanol and plaques were visualized and counted either by immunohistochemistry using BV+ macaque sera as the primary antibody and HRP-conjugated goat-antihuman as the secondary antibody with DAB detection or by staining with crystal violet. IC_50_ values were calculated using logarithmic regression line generated from a semi-log dose–response curve.

### 2.9. Direct Inactivation Using Modified Plaque Reduction Assay

150 PFU of B virus were incubated with 0 μM or 50 μM genistein at 37 °C for 15, 30, 60, 90, and 120 min. At the end of the pre-incubation period, the virus inoculum was removed, diluted 10-fold, then used to infect Veros. All treatments were carried out in duplicate. After 1 h adsorption, cells were overlaid with methylcellulose and incubated at 37 °C for 48 h. Plaques were then counted and the titer calculated.

### 2.10. Virus Yield Assay

Vero cells were grown to confluency in 6-well plates then infected with B virus at a multiplicity of infection (MOI) of 1 and incubated for 1 h to allow for viral attachment. Cells were washed with 1 mL of 50 mM citrate buffer for 2 min to remove unabsorbed viruses, then rinsed twice with sterile PBS. Next, 0–100 μM of genistein or 0.1% DMSO was added to the infected cells and infection was continued for 24 h. The infected cells were harvested by scraping and lysed by three cycles of alternating freezing and thawing. The total infectious virus yields in the presence of each drug concentrations tested were determined by titration on Vero using a standard plaque assay.

### 2.11. Cell-Based ELISA Assay

HFFs and RMFs were grown to confluency in 12-well plates, then infected with 150 PFU of B virus for 1 h in the presence of 0–100 μM genistein or DMSO control. After 1 h absorption, the media was removed and fresh media added. After 48 h, cells/supernatants were collected, serially diluted, and added to Vero grown to confluency in 96-well plates. A standard curve was generated from 10-fold serial dilutions of B virus (1:100–10^9^). After 1 h absorption, the inoculum was removed, fresh media added and infection was continued for 24 h. Next, cells were fixed with 85% acetone, washed three times with borate-buffered salt solution containing 0.05% Tween 20 (BBST), blocked for one hour in ELISA-Blotto (BBST + 2.5% dry milk), and incubated with rabbit anti-gB sera (generated from injection of rabbits with a recombinant B virus gB protein, as previously described [36]) at 1:50 dilution overnight at 4 °C. The primary antibody was removed, cells were washed three times in BBST, incubated with goat anti-rabbit IgG-alkaline phosphatase for 1 h and washed again three times in BBST. For detection, 200 µL of p-nitrophenyl phosphate (pNPP) substrate was added to each well and incubated for 30 min at room temperature. OD values were generated by reading absorbance on an ELISA microplate reader at 405 nm and 490 nm wavelengths. The average OD values were calculated from quadruplicates per experiment. Net OD values were obtained by subtracting the average OD values of the background control (ODbkg) from the average OD values of the test samples (ODts). A titration curve was constructed by plotting the net OD values of the virus standard versus the experimental virus dilution. The titer of the virus was determined from the curve using a cutoff valued that is determined from the background controls. We used the average of the background control OD values plus three standard deviations as the cutoff. The titer of the experimental samples was calculated from the standard curve.

### 2.12. Statistical Analysis

All analyses were performed either in Excel (*t*-test) or using GraphPad Prism 6 software (One-way ANOVA with correction for multiple comparisons—Dunnet’s correction when comparing all groups and Tukey’s correction when comparing all groups to the control group). Deviation from the mean was calculated using the standard deviation (Excel) or the standard error of the mean (GraphPad Prism 6 software).

## 3. Results

### 3.1. Genistein Does Not Exhibit Cytotoxic Effects on Primary Fibroblasts

Primary B virus infection occurs at the epithelial layer of the skin and/or mucosa. Thus, we chose to test genistein’s efficacy in primary human and macaque fibroblasts (HFF and RMF respectively). Genistein, in certain cell types, has been reported to inhibit cellular proliferation via cell cycle arrest, so we first verified that genistein would not have any cytotoxic effects on primary fibroblasts at doses used in this study. We tested the effects of genistein on cell viability, proliferation, metabolic activity, and DNA synthesis. Cell viability was measured at 48 h post-genistein treatment via trypan blue dye exclusion assay. Genistein had no notable effect on cell viability at 48 h post-treatment (Figure 1A, \B). Next we asked if genistein affected cell proliferation. Initially, we assayed cellular division in an asynchronistic population of cells and saw no effect (Figure 1C,D). We then tested if genistein’s effect on cell doubling was masked in the asynchronistic population. Cells were synced in G_0_/G_1_ phase via serum-starvation, then released from cell cycle arrest by the reintroduction of serum with or without the addition of genistein. At all doses, the tested cells showed similar doubling times to the controls (Figure 1C,D). An alternative method of measuring cellular proliferation was performed using an MTS assay, which measures cellular metabolic activity via the cell’s ability to breakdown tetrazolium dye or MTT (3-(4,5-dimethylthiazol-2-yl)-2,5-diphenyltetrazolium bromide) to insoluble formazan. Cells were serum-starved overnight then released from cell cycle arrest with the reintroduction of serum in the presence or absence of genistein. Cells were kept for either 48 or 72 h and then assayed for their ability to reduce MTT. Genistein did not reduce cellular metabolic activity in HFFs; in fact, an increase in metabolic activity was noted, particularly at median doses, which was most apparent after 72 h (Figure 1E). A slight, though insignificant increase in metabolic activity was also repeatedly noted in both cell lines following treatment with DMSO (vehicle control). In RMFs, genistein had a modest inhibitory effect at high doses with a 24% and 36% reduction in metaboloic activity at 50 µM and 100µM, respectively, compared with the DMSO vehicle control (Figure 1F). When compared to untreated cells, the effect was less dramatic, with a 10% reduction in metabolic activity at 50 µM and 26% reduction at 100 µM. Finally, we assayed for the ability of cells to undergo DNA synthesis in the presence of genistein. Cells were brought into cell cycle arrest via serum-starvation, after which serum was added back in the presence or absence of genistein along with a thymidine analog, bromodeoxyuridine (BrdU), that is incorporated into newly synthesized DNA. There was no notable difference in the pattern of BrdU uptake in cells at any genistein concentration tested versus the controls [37]. Collectively, these data support that genistein is non-toxic to primary fibroblasts at the concentrations used in this study.

### 3.2. Genistein Reduces B Virus Spread and Replication in a Dose-Dependent Manner

We used three methods to assay for antiviral properties of genistein against B virus. Initially, we performed high input infections (MOI 5) in the presence or absence of genistein for 24 h, then collected cells/supernatants and back-titered these suspensions on Vero cells via a standard plaque assay. As shown in Figure 2A, genistein significantly reduced plaque formation in a dose-dependent manner. Since this assay required passaging the virus through a second cell type, we next asked if similar results could be obtained by assaying the virus directly into fibroblasts using either cell-ELISA or plaque reduction assays. Cell-ELISA also revealed a clear trend of reduced virus replication with genistein in a dose-dependent manner with an IC_50_ value of 33 μM in HFF (Figure 2B) and 46 μM in RMF (Figure 2C). Next, we performed direct plaque reduction assays. Genistein reduced plaque formation and plaque size in a dose-dependent manner (Figure 2C). Of note, at high doses, virus antigen was rarely detected in the cytoplasm of cells, instead localized to the nucleus. These data suggest that genistein can reduce both productive virus replication and spread, not only in cell-to-cell scenarios but also within the infected cell. Further, our findings support that genistein is effective in restricting B virus infection in both human and macaque cell lines.

Antiviral agents may act directly to inactivate viral particles or, following virus entry into cells, there are several steps that antivirals can target: Immediate-early and early gene synthesis, DNA replication, late gene synthesis, virus assembly and virus budding. Direct plaque reduction assays are an effective means of calculating the reduction in virus titer; however, as shown in Figure 2C, B virus does not produce a clear plaque in primary fibroblasts, making it difficult to obtain reliable quantitative data using this methodology. For these reasons, we performed plaque reduction assays in Veros, as this cell line is sensitive to B virus infection and B virus produces clear plaques in these cells. We verified that genistein could effectively reduce B virus replication in a dose-dependent manner in Veros. Genistein inhibited plaque formation with an IC_50_ value of 56 μM for B virus (data not shown). To examine if genistein could directly inactivate B virus, 50 μM of genistein was pre-incubated with the virus for 15, 30, 60, 90 or 120 min, and then cells were infected with the virus/drug mix. To verify that the effect of genistein was only on the virus and not the cell, the mix was diluted to obtain a 5 μM final concentration of genistein prior to cellular infection. Our results showed that pre-incubation of B virus with 50 μM genistein for up to 2 h prior to infection had no effect on plaque formation, suggesting that genistein does not directly inactivate the virus and its antiviral activity is due to interference in post-infection processes (Figure 3A).

B virus’ life cycle is ordered into three distinct phases of virus gene transcription/translation: Immediate-early (IE), early (E), and late (L), wherein each phase is dependent on the previous. The full replicative cycle is ~12 h in Vero, with IE genes expressed by 2 hpi, E genes expressed between 2–4 hpi and followed by genomic replication, and L gene synthesis up to 8 hpi. From 8–12 hpi, virions are assembled, glycoproteins undergo glycosylation, and virions are released from the infected cell. To look at the effect of genistein at all phases of the viral life cycle, we added 50 μM genistein at 1, 3, 5, 7, and 9 hpi. There was no difference in virus replication at any time point (Figure 3B). Further, there was no statistically significant difference when genistein was added at the same time as B virus (data not shown; Figure 4C), suggesting that genestein does not impact viral entry. To verify that HFF and RMF would behave in a similar fashion, DNA was isolated from cells infected with increasing concentrations of genistein. No significant difference in the total viral DNA levels in infected cells at any concentration was found (Figure 3C,D). Collectively, these data suggest genistein targets a viral event post-genomic replication and L gene synthesis and, further, that this is not a species-dependent mechanism.

### 3.3. Genistein Reduces Effective Dose of Acyclovir and Ganciclovir against B Virus

Acyclovir (AVC) is typically the first antiviral drug administered following human B virus infection; however, if the patient shows symptoms of CNS involvement, ganciclovir (GCV) is typically administered. Like ACV, GCV has significant side effects, particular at high doses and with long-term usage. We examined whether genistein, when administered at 50 μM in combination with ACV or GCV, could enhance inhibition of B virus. First, we tested the IC_50_ dose of genistein with various concentrations of ACV or GCV. Infected cells receiving a combined genistein–ACV/GCV treatment showed marked reduction in plaque formation compared to cells treated with genistein, ACV or GCV alone (Figure 4A,B; Table 1). For example, whereas a 6 μM dose of ACV had little effect on B virus, a co-treatment with genistein at its IC_50_ value of 50 μM resulted in a 75% decrease in plaque formation (Figure 4A; Table 1). GCV is more effective against B virus than ACV, with an IC_50_ value of 16 μM versus 55 μM in Vero (Table 1), and, thus, was tested at a smaller range of concentration. Genistein exhibited the same additive effect when given with GCV (Figure 4B; Table 1). The similar level of plaque reduction was expected since these two nucleoside analogs share similar antiviral actions. The data also showed that increasing the concentration of ACV or GCV produced little changes to the reduction by genistein, suggesting that genistein and these two nucleoside analogs are not necessarily interacting with each other chemically, but may work via independent mechanisms to prevent viral infection. Conversely, we tested the IC_50_ dose of ACV with varying concentrations of genistein and inhibition in plaque formation was virtually identical as that achieved by a cotreatment of ACV and 50 μM genistein (Figure 4C; Table 1). Thus, neither drug can be concluded as an enhancer, but rather that genistein and ACV interact in a synergistic manner resulting in an enhanced antiviral response.

## 4. Discussion

This study tested the hypothesis that the plant flavonoid genistein has antiviral activity against B virus. Using plaque reduction assays and plaque titration assays, we not only showed that genistein can reduce virus replication and spread in a dose-dependent manner, but also that it targets a point in the virus life cycle after DNA replication. Further, we showed that genistein could act synergistically with current antiviral therapies to reduce their effective dose. Finally, we showed that efficacious doses had no cytotoxic effects on primary human and macaque cell lines.

Genistein had IC_50_ values against B virus of 33, 46, and 55 μM in HFF, RMF, and Vero cell lines, respectively. While the antiviral activity of genistein in HFF and RMF cells has not been previously reported on, our data for Vero is congruent with values reported from other groups. The effective concentration of genistein ranges between 12–50 μM [27,28,29,30,32], while toxicity is observed between 25–250 μM, depending on the technique and cell type. The highest value in the toxicity range was reported from Lyu et al. via MTT assay [21]. In our study, cellular toxicity was measured using four complimentary methods and no cytotoxicity was noted in primary fibroblasts at any of the tested doses (12.5–100 μM), however, toxicity was noted above 50 μM in Veros (data not shown). Animal studies of genistein in mice and pigs have shown no adverse effects on growth, reproduction or development [19,31]. Therefore, genistein is an attractive candidate for antiviral therapy.

Our data suggest that genistein is targeting a late-phase event in the life cycle of B virus to block new virion production and prevent viral spread. In our current study, we found that pre-exposure of cells to genistein had no enhanced effect on virus reduction or reduction in viral DNA production, suggesting that genistein targets an intracellular event during the viral life cycle. Productive virus replication proceeds with the organized and temporal expression of three gene family groups—IE, E, and L—with viral DNA replication occurring between E and L phases. Numerous phosphorylation events are involved during each of these phases, which could provide potential targets for a protein kinase inhibitor such as genistein. Some studies have already demonstrated that genistein interferes with late infection by inhibiting phosphorylation of glycoproteins [20,27,30]. In this study, immunohistochemical (IHC) staining for viral antigen revealed a very restricted intracellular distribution with increasing concentrations of genistein. This observation supports the possibility that genistein is interfering with late phase virion assembly and/or trafficking through the endomembrane system. Currently, we are investigating the specific mechanism by which genistein inhibits virion production and viral spread.

The macaque is widely used in the biomedical research community owing, in part, to its high degree of genetic homology to the human. The threat of zoonosis has led to the establishment of specific pathogen-free (SPF) breeding colonies. In the macaque, B virus presents with minimal pathology and macaques in these colonies often show sero-conversion to B virus as they reach sexual maturity [38]. Sporadic sero-conversion, coupled with asymptomatic shedding from infected animals, produces a potentially dangerous situation whereby animal handlers, veterinarians, and researchers can be at risk of exposure. In this study, genistein was equally effective at limiting B virus infection in macaque fibroblasts as human fibroblasts. These observations raise the interesting possibility of using genistein to control outbreaks within these SPF colonies.

Genistein has antiviral activity against several DNA and RNA viruses, though this is the first report of its antiviral activity against B virus in primary human and macaque cell lines [19,20,21,22,23,24,25,26,27,28,29,30,31,32]. Our data suggest that genistein may have therapeutic potential as an augmenter to current antiviral drugs for B virus. Combination therapy has been found to be useful against HSV-1 [39,40,41,42,43]. In most cases, combination therapy against virus infection requires a lower dose of either drug to achieve the same or greater antiviral response. If genistein can synergize with ACV or GCV to reduce the effective dose, then such a combination could greatly reduce negative side effects seen with long-term administration to B virus-infected patients. In a clinical setting, genistein can be provided as a topical ointment to be applied at the site of exposure. This method can enhance the absorption of genistein to the infected site since it bypasses the rapid hepatic metabolism and low bioavailability associated with oral consumption of flavonoids. Furthermore, it can limit toxicity only to the infected area. Topical delivery has been suggested to work well for other flavonoids with demonstrated antiviral activity [44,45]

In conclusion, genistein is a promising new antiviral treatment of human B virus infection and may serve as an augmenter to current, more toxic antiviral treatments.

## Figures and Tables

**Figure 1 viruses-11-00499-f001:**
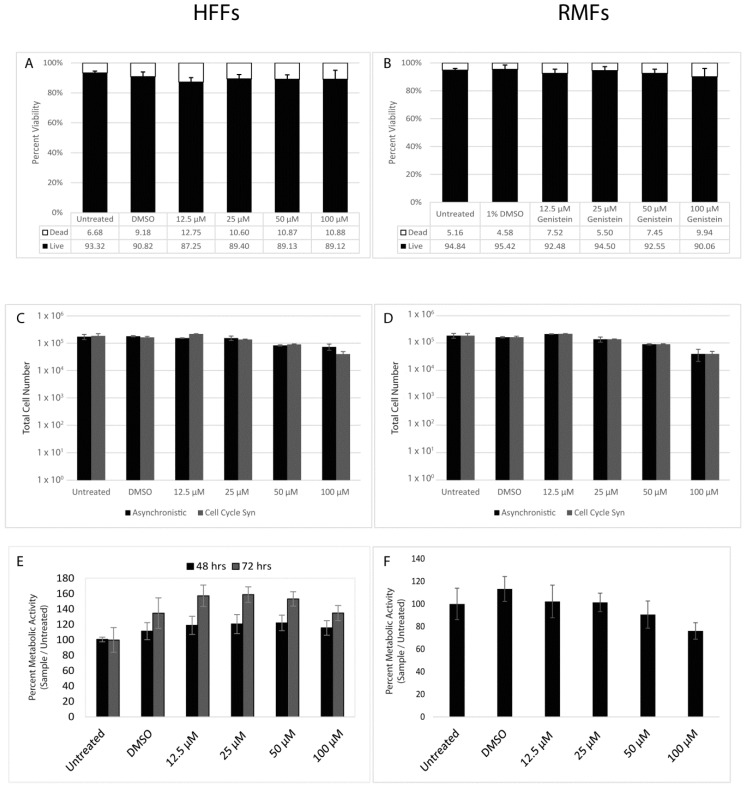
Cytotoxicity of genistein in human foreskin fibroblasts (HFFs) and Rhesus macaque foreskin fibroblasts (RMFs). Cells were left untreated (negative control) or treated with either 0.1% dimethyl sulfoxide (DMSO) (experimental control) or increasing concentrations of genistein. After 48 h, HFFs (**A**) and RMFs (**B**) were collected and assayed for live vs. dead cells via trypan blue dye exclusion. HFFs (**C**) and RMFs (**D**) were kept as an asychronous population or growth was synchronized via overnight serum-starvation prior to genistein treatment. Cells were collected and the total cell numbers counted after 48 h. HFFs (**E**) and RMFs (**F**) were serum-starved overnight, then genistein-treatments were added and cells were incubated for 48 h (**E**) or 72 h (**E**,**F**), and then assayed for metabolic activity via MTS assay. For A–F, *n* = 9, with the statistical analysis performed via one-way ANOVA with Dunnet’s correction for multiple comparisons.

**Figure 2 viruses-11-00499-f002:**
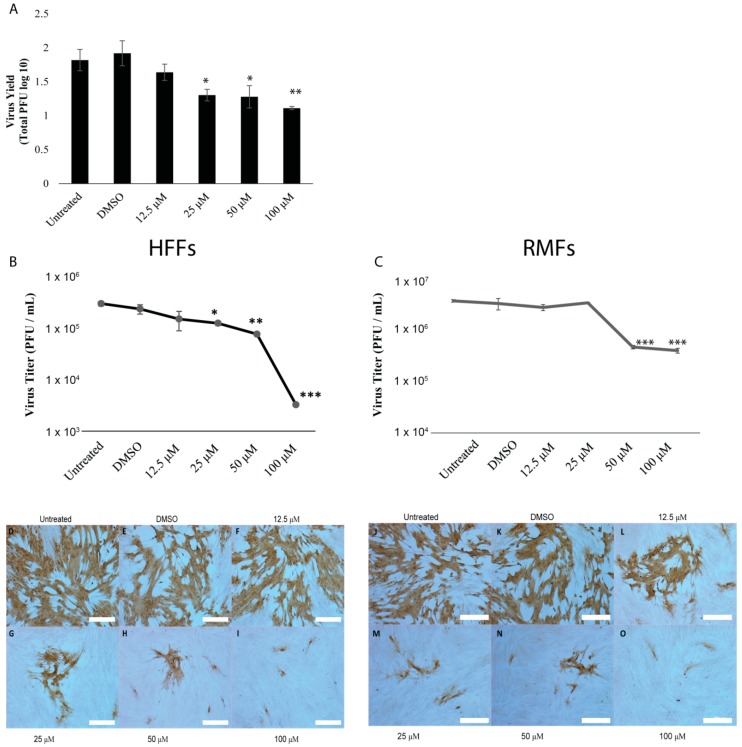
Genistein has antiviral activity against B virus in human and macaque fibroblasts. HFFs and RMFs were left untreated (negative control) or treated with either 1% DMSO (experimental control) or increasing concentrations of genistein and infected with 150 PFU of B virus for 48 h. (**A**) Indirect virus yield assay in HFFs. After 48 h, cells/supernatants were harvested and back-titered on Veros. Data shows total PFU counted in Veros. Cell-based Elisa in HFFs (**B**) and RMFs (**C**). After 48 h, cells were fixed with 100% methanol and assayed via ELISA for amount of virus antigen. An in-assay standard curve was generated to calculate virus titer and IC_50_ values were calculated using a logarithmic regression line. Statistical analysis performed via one-way ANOVA with Dunnet’s correction for multiple comparisons. N = 9. Error bars show standard error of the mean. * *p* = <0.05, ** *p* < 0.01, *** *p* =< 0.001. Direct virus yield assay in HFFs (**D**–**I**) and RMFs (**J**–**O**). After 48 h, cells were fixed with 100% methanol and virus antigen was visualized via IHC with DAB detection. Scale bars shown at 200 µM.3.3. Genistein Targets B Virus Post-Viral Entry and Genomic Replication.

**Figure 3 viruses-11-00499-f003:**
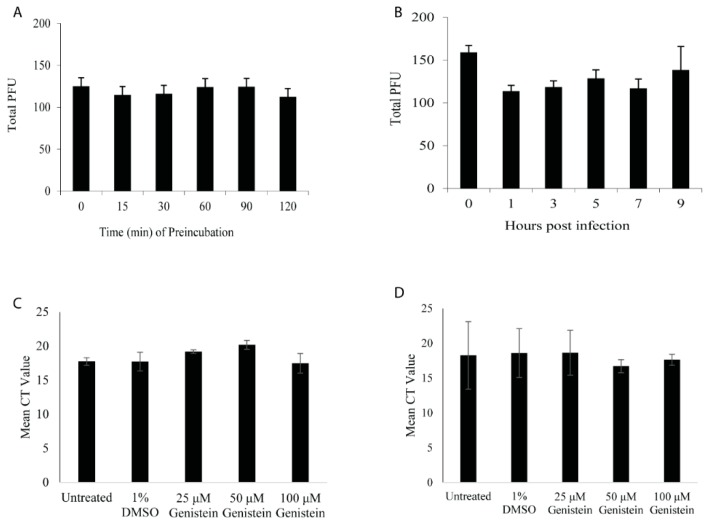
Genistein does not directly inactivate B virus; instead, it inhibits the virus after virus replication. (**A**) Direct inactivation assay. Veros were infected with 150 PFU of B virus following pre-incubation of B virus with 50 µM genistein for 0–120 min. A standard plaque assay was performed and the total PFU counted. (**B**) Plaque reduction assay. Veros were infected with 150 PFU of B virus and 50 µM of genistein was added at various times post-infection. A standard plaque assay was performed and the total PFU counted. RT-PCR results for total genomic virus post-48 h infection of HFFs (**C**) or RMFs (**D**) following B virus infection in the presence of increasing concentrations of genistein or 1% DMSO (control).

**Figure 4 viruses-11-00499-f004:**
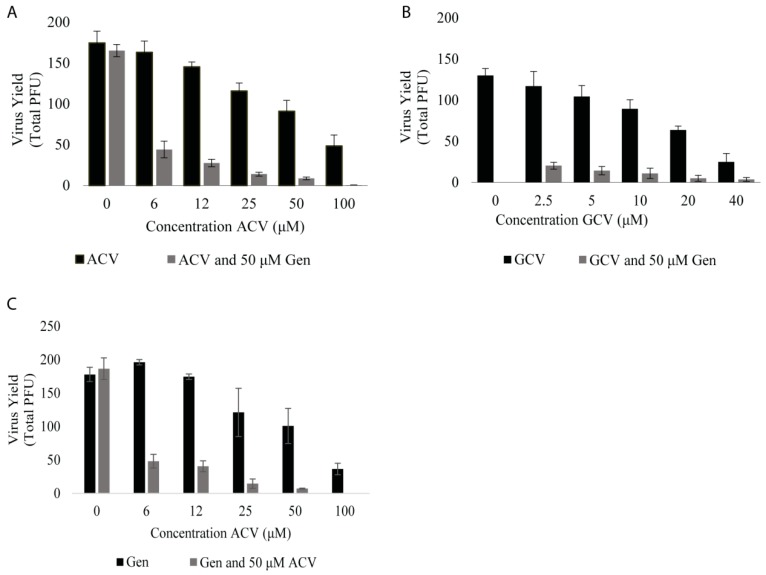
Genistein acts synergistically with antiviral agents to reduce effective dose. (**A**) and (**B**) show the results of virus yield assays, wherein Vero cells were infected with 150 PFU of B virus and 50 μM of genistein with increasing concentrations of ACV (**A**) or GCV (**B**). (**C**) Vero cells were infected with 150 PFU of B virus and 50 μM ACV with increasing concentrations of genistein. Total PFU was determined by standard plaque assay.

**Table 1 viruses-11-00499-t001:** Calculated IC_50_ Values against B Virus.

Agent	Concentration (µM)
Genistein	56
Acyclovir	55
Acyclovir + Genistein (50 µM)	3.9
Ganciclovir	16
Ganciclovir + Genistein (50 µM)	0.002

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
