# Peer review of "Genistein Has Antiviral Activity against Herpes B Virus and Acts Synergistically with Antiviral Treatments to Reduce Effective Dose"

_viruses, 2019, doi:10.3390/v11060499_

Reviewer 1 Report

The paper would be greatly strengthened if additional experiments to determine the mechanism of action were included.  EM of infected cells +/-drug could be informative as would immunoblots of the glycoproteins since it appears to be effective even at late times.  Is release inhibited?

Line 131, what does "insert mods." mean?

Author Response

Reviewer 1:

Thank you very much for your through and thoughtful review of our manuscript.  

Point 1:The paper would be greatly strengthened if additional experiments to determine the mechanism of action were included.  EM of infected cells +/-drug could be informative as would immunoblots of the glycoproteins since it appears to be effective even at late times.  Is release inhibited?

Response 1: We appreciate your suggestion of additional experiments to determine genistein’s mechanism of action. We would be happy to do these types of experiments had we the time, but the short turnaround and rigors of BSL4 work precludes that possibility. EM of cells post infection with and without genistein is an excellent idea and this experiment will be included in our continued study of genistein’s mechanism of action against B virus infection. Thank you for your interest in our continued work.

Point 2: Line 131, what does "insert mods." mean?

 Response 2: We apologize for the inclusion of “insert mods” on line 131 in the Methods section. This was a leftover from revision and we have deleted it. Thank you very much for catching that.

Reviewer 2 Report

This is a well written and well presented study on the effect of genistein on herpes B virus replication but I see two interpretation and/or control issues that need to be addressed before publication.

First, In Figure1F they say there is no effect on metabolic activity yet there clearly is an effect on the RMFs. I have a pet peeve when it comes to authors stating there is no effect yet clearly the graph shows one. The proper control for the drug is DMSO and when comparing to that control the drug tends to reduce the activity. At 50 and 100 uM there is clearly a reduction in metabolic activity of 20 and 30% in the RMFs. Unless they want to provide stats on this to prove me wrong then they should state there is some modest effect of the drug on this cell line.

Second, from my reading figure 3B does not have a time 0 control with the drug added at the same time as infection. If drug is present at time 0 then that must be clarified. If it is not present then the expt needs to be done with the drug at time 0. If we take the data from 1 hr to 9 hrs it appears the drug is less effective over time (except for time 7 hrs which is odd). There may not be much of an effect but lacking the time 0 with drug is an issue.

other issues:

1) no references for the drugs effect on other viruses except for Porcine virus-these should be added

see intro "Genistein has been shown to have antiviral properties against several herpesviruses including HSV-1, cytomegalovirus, and bovine herpesvirus-1, as well as non-herpesviruses including SV40, human papilloma virus, and porcine reproductive and respiratory syndrome virus [20]

2) not the discussion is truncated at line 376 and ends mid-sentence???

3) error in Figure 3 "at genisten" and no concentration stated

Author Response

Reviewer 2:

Thank you very much for your through and thoughtful review of our manuscript. We are pleased that you find it a well written and well-presented study. We have taken your concerns and revisions into account and feel that our manuscript is greatly improved by these changes.

Point 1: First, In Figure1F they say there is no effect on metabolic activity yet there clearly is an effect on the RMFs. I have a pet peeve when it comes to authors stating there is no effect yet clearly the graph shows one. The proper control for the drug is DMSO and when comparing to that control the drug tends to reduce the activity. At 50 and 100 uM there is clearly a reduction in metabolic activity of 20 and 30% in the RMFs. Unless they want to provide stats on this to prove me wrong then they should state there is some modest effect of the drug on this cell line.

Response 1: We apologize for our erroneous omission of a discussion of the effect of genistein on the metabolic activity of RMFs at high doses. You are quite right and we have changed the manuscript to reflect this in lines 214 - 220. Thank you for catching this oversight.

Point 2: Second, from my reading figure 3B does not have a time 0 control with the drug added at the same time as infection. If drug is present at time 0 then that must be clarified. If it is not present then the expt needs to be done with the drug at time 0. If we take the data from 1 hr to 9 hrs it appears the drug is less effective over time (except for time 7 hrs which is odd). There may not be much of an effect but lacking the time 0 with drug is an issue.

Response 2: In Figure 3B we sought to examine the effect of genistein on productive virus replication when administered at different times post infection. In this set of experiments we treated cells with 50 µM genistein starting at the post adsorption time point, 1 hour post infection. We did not include the control of co-administration of drug and virus within this set of experiments. However, in our initial experiments to determine genistien’s IC50 value we did add genistein directly with B virus at the time of infection. In the original manuscript we elected to state this experiment as ‘data not shown’ as this experiment was essentially redone in Figure 4C where we examined the synergistic effects of genistein and acyclovir. When comparing Figure 3B with Figure 4C, co-administration of 50 µM of genistein with B virus results in 101 PFUs which is insignificantly different then what we see if we add the drug at 1 hour post infection. We have changed the manuscript to add a discussion of this in the Results section lines 286 - 288. This is an important point to show that the drug is not inhibiting viral entry during co-administration. We would be happy to repeat these experiments with the in-experiment control had we the time, but the short turnaround and rigors of BSL4 work precludes that possibility We thank you very much for bringing this to our attention and hope that the inclusion of our interpretation meets with your satisfaction.

Point 3: no references for the drugs effect on other viruses except for Porcine virus-these should be added see intro "Genistein has been shown to have antiviral properties against several herpesviruses including HSV-1, cytomegalovirus, and bovine herpesvirus-1, as well as non-herpesviruses including SV40, human papilloma virus, and porcine reproductive and respiratory syndrome virus [20]

Response 3: We apologize for the exclusion of necessary references in the Introduction for lines 77 – 80 and in the Discussion lines 373 and 374. This was an oversight on our part and we appreciate you catching the mistake. The proper references have been added.

Point 4: not the discussion is truncated at line 376 and ends mid-sentence???

Response 4: Thank you for bringing to our attention the final truncated sentence in the Discussion, line 386. It has been deleted.

Point 5: error in Figure 3 "at genisten" and no concentration stated

Response 5: Again, thank you for catching the error in the legend for Figure 3. We have added in the concentration of genistein used in figures 3 A and B (lines 297 and 298) and have changed “at genistein” to “of genistein” on line 299.

Reviewer 3 Report

The Authors explored in this work the activity of genistein as potent anti-Herpes B virus agent. The biological studies are properly described and supported by experimental evidence. 

The manuscript is well described and references are adequate and updated,

Some minor revisions should be addressed:

1- In the introduction, please cite the more recent references near citation [20]:

a) Arabyan E, et al. Antiviral Res. 2018;156:128-137

b) Ozçelik B, et al. Pharm Biol. 2011;49(4):396-402

2- page 5, row 202: modify "effected" into "affected"

  page 8, row 275: "phase it dependent" into "phase is dependent"

Thus, I believe this manuscript is of interest for scientists working in this field, and therefore I recommend its publication in Viruses.

Author Response

Reviewer 3: 

Thank you very much for your through and thoughtful review of our manuscript. We are very pleased that you find our work of interest to the field.

Point 1: In the introduction, please cite the more recent references near citation [20]:

a) Arabyan E, et al. Antiviral Res. 2018;156:128-137

b) Ozçelik B, et al. Pharm Biol. 2011;49(4):396-402

Response 1: We apologize for the exclusion of necessary references in the Introduction for lines 77 – 80 and in the Discussion lines 373 and 374. This was an oversight on our part and we appreciate you catching the mistake. The proper references have been added.

Point 2: page 5, row 202: modify "effected" into "affected".  page 8, row 275: "phase it dependent" into "phase is dependent"

Response 2: Thank you for bringing to our attention grammatical errors on page 5, line 204 and page 8, line 281. “effected” has been changed to “affected” on line 204 and “phase it dependent” has been changed to “phase is dependent” on line 281.

Round  2

Reviewer 2 Report

Good to go, the changes made are good enough for publication.